# Enhanced insulin signalling ameliorates C9orf72 hexanucleotide repeat expansion toxicity in *Drosophila*

Magda L Atilano[1,2], Sebastian Grönke[3], Teresa Niccoli[1,2], Liam Kempthorne[2,4], Oliver Hahn[3], Javier Morón-Oset[3], Oliver Hendrich[3], Miranda Dyson[1,2], Mirjam Lisette Adams[1,2], Alexander Hull[1], Marie-Therese Salcher-Konrad[2,4], Amy Monaghan[5], Magda Bictash[5], Idoia Glaria[2,4], Adrian M Isaacs[2,4*], Linda Partridge[1,3*]

[1]Department of Genetics, Evolution and Environment, Institute of Healthy Ageing, London, United Kingdom; [2]UK Dementia Research Institute at UCL, London, United Kingdom; [3]Max Planck Institute for Biology of Ageing, Cologne, Germany; [4]Department of Neurodegenerative Disease, UCL Institute of Neurology, London, United Kingdom; [5]Alzheimer's Research United Kingdom UCL Drug Discovery Institute, University College London, London, United Kingdom

**\*For correspondence:**
a.isaacs@ucl.ac.uk (AMI);
Linda.Partridge@age.mpg.de (LP)

**Competing interests:** The authors declare that no competing interests exist.

**Abstract** G4C2 repeat expansions within the *C9orf72* gene are the most common genetic cause of amyotrophic lateral sclerosis (ALS) and frontotemporal dementia (FTD). The repeats undergo repeat-associated non-ATG translation to generate toxic dipeptide repeat proteins. Here, we show that insulin/IGF signalling is reduced in fly models of *C9orf72* repeat expansion using RNA sequencing of adult brain. We further demonstrate that activation of insulin/IGF signalling can mitigate multiple neurodegenerative phenotypes in flies expressing either expanded G4C2 repeats or the toxic dipeptide repeat protein poly-GR. Levels of poly-GR are reduced when components of the insulin/IGF signalling pathway are genetically activated in the diseased flies, suggesting a mechanism of rescue. Modulating insulin signalling in mammalian cells also lowers poly-GR levels. Remarkably, systemic injection of insulin improves the survival of flies expressing G4C2 repeats. Overall, our data suggest that modulation of insulin/IGF signalling could be an effective therapeutic approach against *C9orf72* ALS/FTD.

## Introduction

Amyotrophic lateral sclerosis (ALS) and frontotemporal dementia (FTD) are fatal neurodegenerative diseases. The most common genetic cause is a hexanucleotide G4C2 repeat expansion in the first intron of the *C9orf72* gene (*DeJesus-Hernandez et al., 2011*; *Renton et al., 2011*; *Gijselinck et al., 2012*). Possible mechanisms for the expansion-related neurodegeneration include (i) haploinsufficiency of the *C9orf72* gene; (ii) transcription of the repeats in the sense and antisense direction with accumulation of RNA foci that sequester RNA binding proteins; (iii) production of toxic dipeptide-repeat proteins (DPRs) through repeat-associated non-AUG (RAN) translation (*Gijselinck et al., 2012*; *Balendra and Isaacs, 2018*; *Zu et al., 2018*).

From the five DPRs produced (poly-GA, poly-GR, poly-GP, poly-PA, and poly-PR), the arginine-containing dipeptides poly-GR and poly-PR are highly toxic in cell lines, cultured neurons, *Drosophila*, and mice (*Mizielinska et al., 2014*; *Wen et al., 2014*; *Zhang et al., 2018b*; *Zhang et al., 2019*; *Moens et al., 2019*). Several mechanisms contribute to DPR-induced toxicity, including impaired translation, nucleolar stress, DNA damage, impaired nucleocytoplasmic transport, and altered stress granule dynamics (*Kwon et al., 2014*; *Freibaum et al., 2015*; *Jovičić et al., 2015*; *Kanekura et al.,*

*2016*; *Lee et al., 2016*; *Lopez-Gonzalez et al., 2016*; *Boeynaems et al., 2017*; *Zhang et al., 2018a*; *Moens et al., 2019*).

Insulin/insulin-like growth factor (IGF) signals through a highly conserved pathway that regulates a multitude of processes such as protein synthesis, cell growth, and cell differentiation in both vertebrates and invertebrates (*Barbieri et al., 2003*; *Broughton and Partridge, 2009*). In vertebrates, insulin and IGFs are subspecialized into systems with overlapping but distinct biological functions, while in invertebrates there is a single insulin-like system that has the dual function of insulin/IGF signalling.

Activation of the insulin receptor (InR)/IGF-1 receptor through binding insulin/IGF-1 triggers the recruitment of an insulin receptor substrate, which in turn activates PI3-kinase and subsequently 3-phosphoinositide-dependent protein kinase 1 (PDK1). PDK1 can regulate translation via Akt and S6K. PDK1 activates Akt, which then activates mechanistic target of rapamycin (mTOR), which phosphorylates translation initiation factor 4E-binding protein (4E-BP), subsequently releasing its inhibition of the translation initiation factor 4E (eIF4E). PDK1 can also activate translation initiation factor 4B (eIF4B) through ribosomal protein S6 kinase (S6K) phosphorylation. Consequently, translation initiation is increased, protein synthesis is up-regulated, and the proteostatic machinery is also up-regulated (*Brogiolo et al., 2001*; *Wullschleger et al., 2006*; *Sonenberg and Hinnebusch, 2009*; *Martina et al., 2012*; *Minard et al., 2016*).

IGF-1 is an important neurotrophin for the maintenance and survival of motor neurons, and in vivo studies in mouse models of SOD1-ALS have suggested that IGF-1 and IGF-2 have therapeutic efficacy (*Kaspar et al., 2003*; *Allodi et al., 2016*). However, the connection between insulin/IGF signalling and C9ALS/FTD is not yet clear.

Here, we present evidence that expression of (G4C2)36 in adult *Drosophila* neurons leads to a decrease in the levels of the insulin receptor ligands *dilp*2, *dilp*3, and *dilp*5. Furthermore, overexpression of an active form of the single fly insulin/IGF receptor InR in neurons, or insulin treatment, partially rescued the toxicity associated with poly-GR expression in *C9orf72* repeat fly models. Increased insulin/IGF signalling lowered the level of poly-GR in both fly neurons and mammalian cells, identifying a mechanism of rescue of poly-GR toxicity. Our findings indicate that enhanced insulin/IGF signalling may provide a potential therapeutic target to ameliorate the toxic effects of the *C9orf72* repeat expansion.

## Results

### Insulin signalling is down-regulated in flies expressing expanded *C9orf72* repeats

To identify disease-specific gene expression patterns associated with poly-GR, we performed RNA-seq on heads of adult flies expressing ATG-driven GR100 specifically in neurons, using the Elav-GS RU486-inducible driver. RNA-seq was performed three days after induction of GR100 expression, in order to identify early changes prior to overt neurodegeneration. These flies showed strong alterations in their transcriptome when compared with the control line. We identified 2754 genes significantly differentially regulated (adjusted p<0.05) (*Figure 1—figure supplement 1A* and GEO: GSE151826). To gain insight into the potential function of the differentially expressed genes, we performed gene ontology (GO) enrichment analysis using TopGO on the 2754 up- and down-regulated genes (*Figure 1A*). Flies expressing poly-GR100 showed altered expression of genes involved in pathways previously implicated in *C9orf72* pathology, including translation, DNA damage and repair, proteasome, and RNA metabolism (*Figure 1A* and *Figure 1—figure supplement 1B*). Interestingly, neuropeptide hormone activity was the most enriched category of down-regulated genes (*Figure 1A*). Among these neuropeptides, *Drosophila* insulin-like peptides (dilps) 2, 3, and 5 were highly down-regulated (3.7-, 9.5-, and 3.2-fold change, respectively), adjusted p value<0.05 – *Figure 1—figure supplement 1A*. RT-PCR analysis of brains of flies expressing poly-GR confirmed the lowered expression of *dilp 2, 3*, and 5 (*Figure 1B*). To test whether these *dilps* were also down-regulated in a GGGGCC repeat model that generates poly-GR via RAN translation, we performed RT-PCR in flies expressing (G4C2)36. Lower expression of *dilps* was also observed in this model, although the expression change in the *dilp5* gene did not reach statistical significance (*Figure 1—figure supplement 2*).

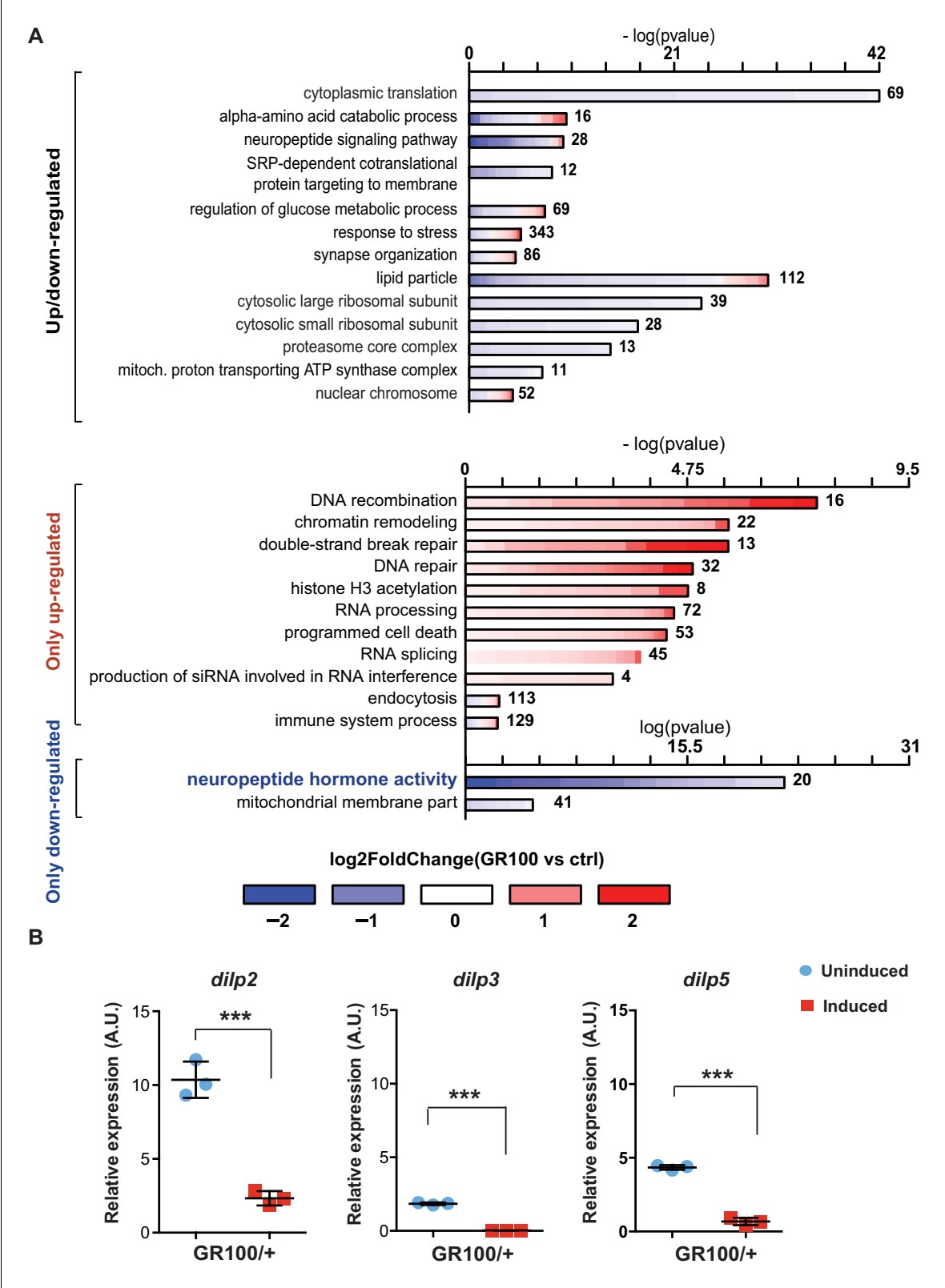

**Figure 1.** Insulin signalling is down-regulated in flies expressing *C9orf72* repeats. (**A**) Gene ontology enrichment of genes with altered expression when GR100 was expressed in neurons. In the top graph, bars represent enrichment of up- and down-regulated genes. In the bottom graph, upper bars represent only up-regulated genes, lower bars represent only down-regulated genes. Lengths of bars represent negative log-transformed, adjusted p-values for Fisher's exact enrichment test. Bar colour indicates log2-fold changes between GR100 and control per gene. Neuropeptide/hormone

*Figure 1 continued on next page*

*Figure 1 continued*

activity genes were down-regulated. (**B**) Quantitative RT-PCR analysis of dilp 2, 3, and 5 normalized against tubulin in fly heads expressing GR100 in neurons. Data was assessed by t-test and presented as mean ± SD, n = 3; dilp2: p=0.0004; dilp3: p<0.0001, dilp5: p<0.0001. Genotypes: (A) w; UAS-GR100/+; ElavGS/+ (GR100), w; +; ElavGS/+ (ctrl) and (B) w; UAS-GR100/+; ElavGS/+.

The online version of this article includes the following source data and figure supplement(s) for figure 1:

**Source data 1.** Source data pertaining to *Figure 1A*.
**Source data 2.** Source data pertaining to *Figure 1B*.
**Figure supplement 1.** Expression of GR100 in fly neurons induces strong perturbation of the transcriptome.
**Figure supplement 2.** Insulin-like peptides 2, 3, and 5 are down-regulated in flies expressing (G4C2)36.
**Figure supplement 2—source data 1.** Excel sheet containing source data pertaining to *Figure 1—figure supplement 2*.
**Figure supplement 3.** Expression of GR100 in fly neurons does not induce loss of IPCs.
**Figure supplement 3—source data 1.** Excel sheet containing source data pertaining to *Figure 1—figure supplement 3C*.
**Figure supplement 4.** Insulin pathway activity is down-regulated in flies expressing (G4C2)36.
**Figure supplement 4—source data 1.** Source data pertaining to *Figure 1—figure supplement 4B*.

Dilps 2, 3, and 5 are neuropeptides secreted by insulin-producing cells (IPCs) in the *Drosophila* brain (*Brogiolo et al., 2001*; *Ikeya et al., 2002*). To investigate whether reduced expression of *dilps* was simply due to the death of IPCs, we imaged brains of flies expressing poly-GR in all neurons. Dilp2 immunostaining showed no alteration in the number of IPCs (*Figure 1—figure supplement 3A and B*). However, consistent with our RNA-seq data, Dilp2 protein levels were reduced in IPCs expressing poly-GR (*Figure 1—figure supplement 3C*). In addition, expression of poly-GR specifically in IPCs using the dilp3-Gal4 driver was not sufficient to induce neuronal cell death (*Figure 1—figure supplement 3D*). Reduction of *dilp2*, *dilp3*, and *dilp5* expression in GR100 flies was therefore not due to loss of IPCs.

*Dilps* signal via a single insulin/IGF receptor (InR), through which they regulate the main signalling pathways that modulate 4E-BP1 phosphorylation (*Figure 1—figure supplement 4A*), a well-described read-out of insulin signalling in *Drosophila* (*Bhandari et al., 2001*; *Wang et al., 2003*; *Blancquaert et al., 2010*). To assess insulin/IGF pathway activity, we measured the phosphorylation state of 4E-BP1 in (G4C2)36 flies, and found that they had significantly decreased ratio of phosphorylated 4E-BP1 to the non-phosphorylated form, indicative of reduced insulin signalling (*Figure 1—figure supplement 4B*). Co-expressing a constitutively active insulin receptor (InR[Active]) significantly increased this ratio, indicating a rescue of insulin/IGF signalling immediately downstream of the dilps (*Figure 1—figure supplement 4B*). Taken together, these data indicate a reduction in insulin/IGF pathway activity in flies expressing either G4C2 repeats or poly-GR as a result of a reduction in expression of the *dilps.*

## Activation of insulin signalling reduces G4C2 repeat toxicity in vivo

We next asked whether restoring insulin signalling in neurons could ameliorate G4C2 repeat toxicity. To assess this, we monitored the survival of flies co-expressing (G4C2)36 and InR[Active], specifically in neurons, and found that their lifespan was significantly extended (p<0.001) (*Figure 2A*). In contrast, reducing insulin signalling through the expression of dominant-negative InR (*InR[DN]*) significantly reduced lifespan (p=0.027) (*Figure 2A*). Activating insulin signalling in neurons reduced lifespan of wild-type flies (InR[Active]p<0.0001), as previously reported (*Ismail et al., 2015*), whilst its reduction led to increased lifespan (p=0.035) (*Figure 2B*), again as previously reported (*Augustin et al., 2018*). These observations indicate that increasing insulin/IGF signalling specifically suppresses *C9orf72* repeat toxicity.

Induction of the neuron-specific driver requires flies to ingest RU486. To exclude the possibility that the rescue effect was a consequence of decreased fly feeding, and therefore reduced induction of the (G4C2)36 transgene, we measured food intake. There was no significant difference between the amount ingested across the different experimental groups (*Figure 2—figure supplement 1A*). To rule out the possibility that expression of either InR[Active] or InR[DN] had a direct effect on the transcription of the G4C2 transgene, we measured the transcript repeat levels by RNA dot blot analysis in flies ubiquitously expressing the two constructs. Neither expression of InR[Active] or InR[DN] altered the G4C2 transcript levels (*Figure 2—figure supplement 1B*). To further investigate whether expression of InR[Active] or InR[DN] indirectly affected the inducible protein expression system, we measured

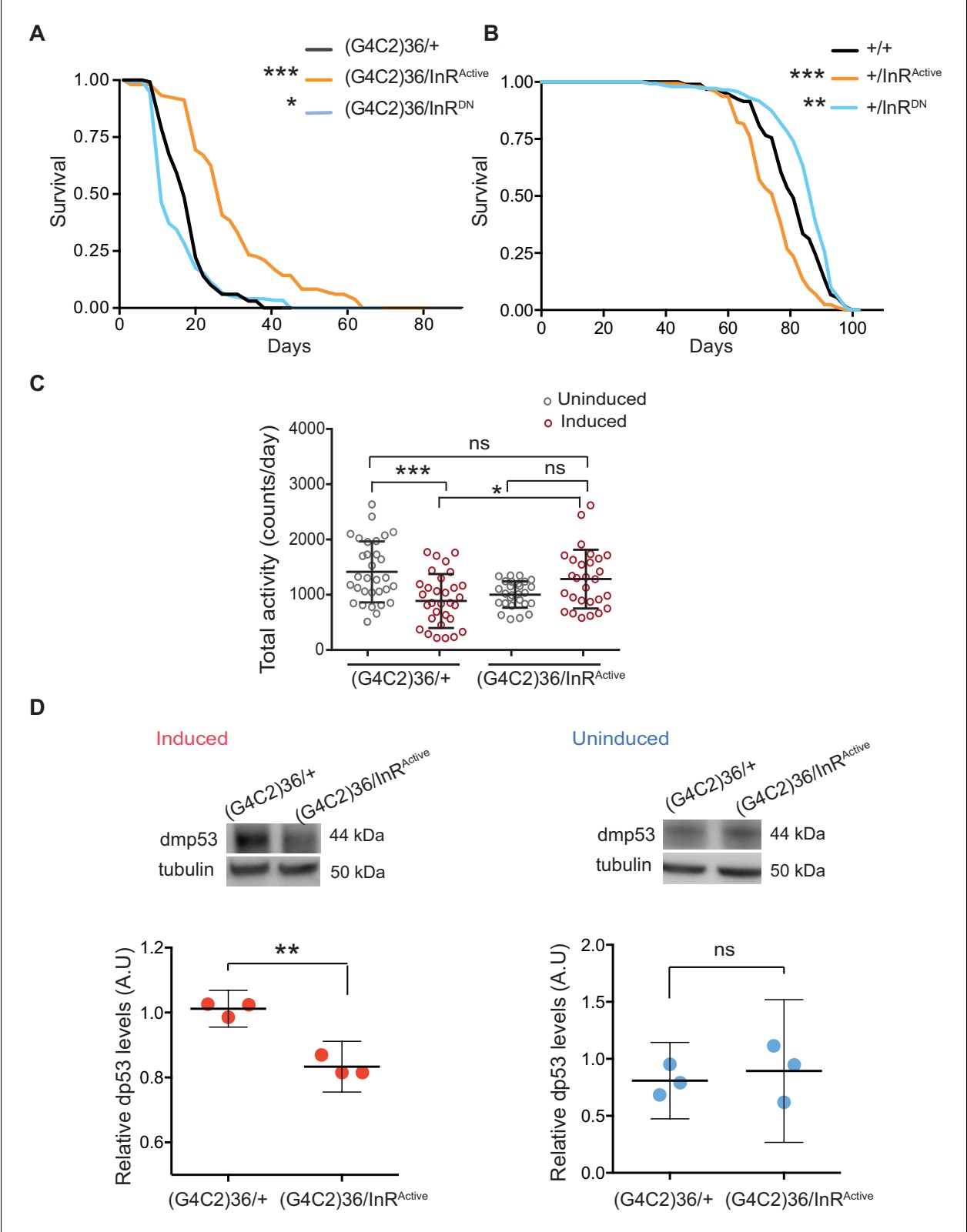

**Figure 2.** Activation of insulin signalling reduces G4C2 repeat toxicity in vivo. (**A**) Lifespan of flies (n = 150) expressing (G4C2)36 or co-expressing InR constructs (InR^Active, InR^DN) in neurons. Lifespan was significantly extended in (G4C2)36 disease flies co-expressing InR^Active compared with (G4C2)36 expressing flies (***p=$2.8 \times 10^{-21}$ – log-rank test) and decreased when co-expressed with InR^DN (*p=0.027). (**B**) Lifespan of wild-type flies (n = 150) expressing InR^Active or InR^DN in neurons. Lifespan was significantly reduced in flies expressing InR^Active compared with control flies (***p=$2.64 \times 10^{-6}$ –
*Figure 2 continued on next page*

Figure 2 continued

log-rank test) and increased in flies expressing InR$^{DN}$ (**p=0.0035). (C) Total activity of flies expressing (G4C2)36 in neurons was significantly reduced compared with uninduced control flies (***p=0.0003). (G4C2)36 flies co-expressing InR$^{Active}$ showed increased activity (*p=0.018) compared with flies expressing (G4C2)36 alone (two-away ANOVA followed by Holm-Sidak's comparison test). Data are presented as mean with SD (n = 30 per genotype). (D) Flies expressing (G4C2)36 alone had significantly increased levels of p53 compared with flies expressing InR$^{active}$ (**p=0.0014, t-test). Data are presented as mean ±95% confidence intervals, n = 3. Genotypes (A) w; UAS-(G4C2)36/+; ElavGS/+, w; UAS-(G4C2)36/UAS-InR$^{Active}$; ElavGS/+, w; UAS-(G4C2)36/UAS-InR$^{DN}$; ElavGS/+. (B) w; ElavGS/+, w; +/UAS-InR$^{Active}$; ElavGS/+, w; +/UAS-InR$^{DN}$; ElavGS/+. (C and D) w; UAS-(G4C2)36/+; ElavGS/+, w; UAS-(G4C2)36/UAS-InR$^{Active}$; ElavGS/+.

The online version of this article includes the following source data and figure supplement(s) for figure 2:

**Source data 1.** Source data pertaining to *Figure 2A*.
**Source data 2.** Source data pertaining to *Figure 2B*.
**Source data 3.** Source data pertaining to *Figure 2C*.
**Source data 4.** Source data pertaining to *Figure 2D*.
**Figure supplement 1.** Expression of InR construct does not affect fly feeding or ElavGS expression system.
**Figure supplement 1—source data 1.** Excel sheet containing source data pertaining to *Figure 2—figure supplement 1A*.
**Figure supplement 1—source data 2.** Source data associated to *Figure 2—figure supplement 1B*.
**Figure supplement 1—source data 3.** Source data pertaining to *Figure 2—figure supplement 1C*.
**Figure supplement 1—source data 4.** Excel sheet containing source data pertaining to *Figure 2—figure supplement 1D and E*.

levels of GFP driven by the Elav-GS RU486-inducible driver and found that they were unaltered in neuronal cells expressing InR$^{Active}$ or InR$^{DN}$ (*Figure 2—figure supplement 1C*). Activation of insulin/IGF signalling therefore ameliorated *C9orf72* repeat toxicity, rather than simply reducing expression of the (G4C2)36 transgene.

To confirm that increasing insulin signalling could also ameliorate a second, distinct (G4C2)36-induced neuronal phenotype, we recorded motor activity and day sleep using a *Drosophila* activity monitor system. Flies expressing (G4C2)36 in adult neurons exhibited decreased locomotor activity (p=0.0003) and extended sleep periods during the day and night (*Figure 2C* and *Figure 2—figure supplement 1D and E*), and these phenotypes were abolished by expression of InR$^{Active}$ (*Figure 2C* and *Figure 2—figure supplement 1D and E*).

We next determined whether activation of insulin/IGF signalling also rescued a molecular signature of (G4C2)36 toxicity. Increased levels of p53 have been observed in *C9orf72* patient iPSC-neurons and fly models, and have been suggested to be a downstream marker of repeat-induced toxicity (*Lopez-Gonzalez et al., 2016*; *Lopez-Gonzalez et al., 2019*). We therefore tested if co-expression of InR$^{active}$ in (G4C2)36 flies was associated with decreased levels of p53 and found that it led to a significant reduction (*Figure 2D*), whereas no difference was observed in uninduced flies (*Figure 2D*). These results show that increased insulin/IGF signalling can ameliorate multiple read-outs of G4C2 repeat-induced toxicity.

## Activation of insulin/IGF signalling reduces poly-GR toxicity in vivo via InR/PI3K/Akt

As toxicity in the (G4C2)36 flies is mediated by poly-GR (*Mizielinska et al., 2014*), and increased p53 has been suggested to be driven by poly-GR (*Lopez-Gonzalez et al., 2016Lopez-Gonzalez et al., 2019*), we next assessed whether increased insulin/IGF signalling could specifically rescue poly-GR toxicity.

We took advantage of the rough eye phenotype and degeneration of eye tissue caused by expression of 36 poly-GR repeats (GR36) in the *Drosophila* eye (*Mizielinska et al., 2014*). To examine whether this pathology could be ameliorated by increased insulin/IGF signalling, we co-expressed GR36 with either InR$^{Active}$ or InR$^{DN}$ (*Figure 3A*). As previously reported (*Mizielinska et al., 2014*), ectopic expression of GR36 resulted in a mild rough eye and decreased eye size (*Figure 3A*). Co-expression of GR36 with InR$^{Active}$ increased the size and decreased the roughness of the eyes, whereas co-expression with InR$^{DN}$ exacerbated the rough eye phenotype and further decreased eye size (*Figure 3B*). Although insulin/IGF signalling also influenced eye cell growth in flies not expressing the dipeptide repeats (*Figure 3B*), the effect in diseased flies was much larger (p<0.0001, two-way ANOVA), indicating a specific additional interaction of insulin signalling with *C9orf72* repeat-induced toxicity. Expression of dilp2 had no effect (*Figure 3C*), likely

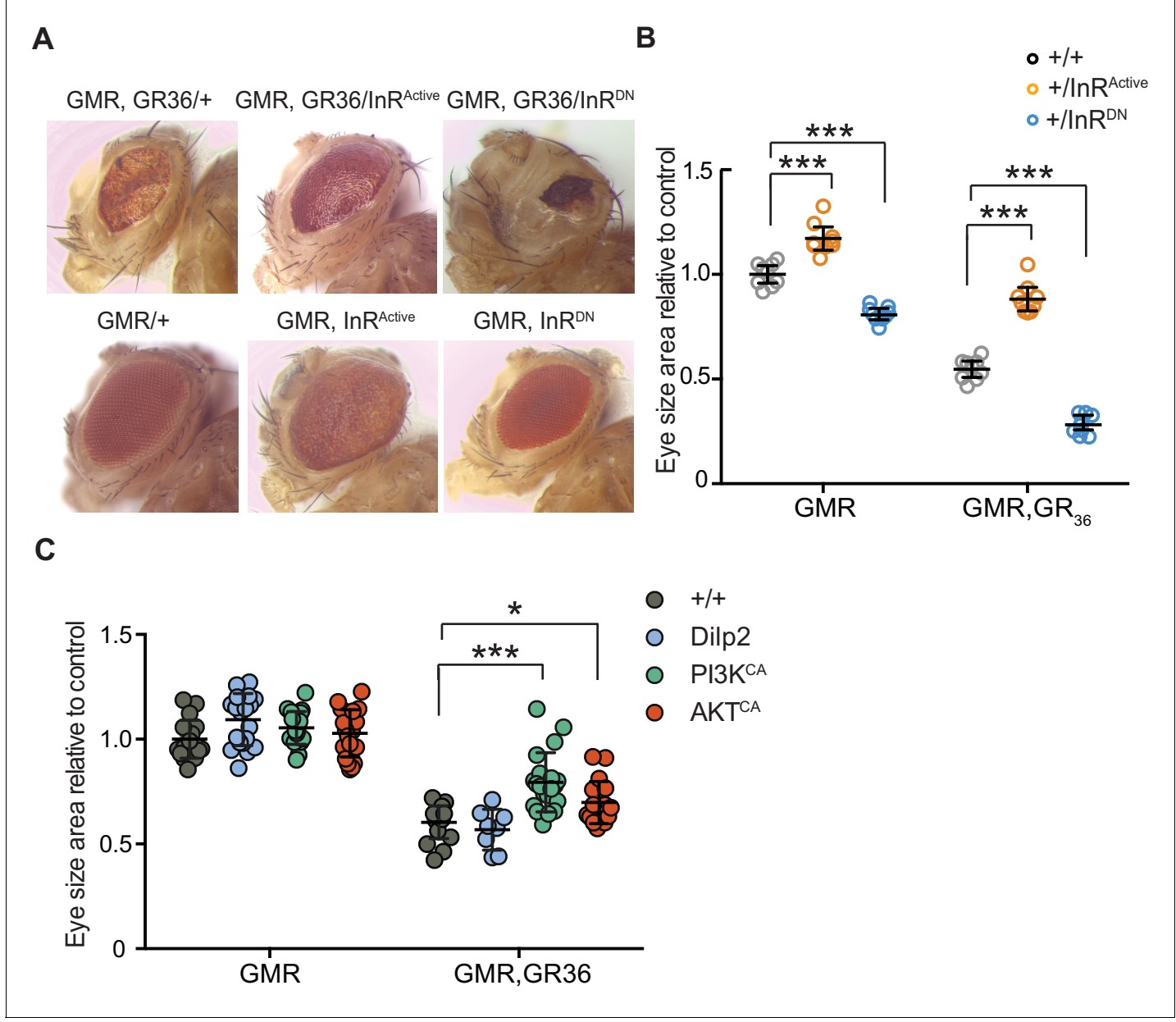

**Figure 3.** Activation of insulin signalling reduces poly-GR toxicity via InR/PI3K/Akt. (**A**) Stereomicroscopy images of representative 2-day-old adult *Drosophila* eyes expressing InR$^{Active}$ or InR$^{DN}$ using the GMR-GAL4 driver (bottom panel) or co-expressing both GR$_{36}$ and InR constructs (top panel). (**B**) Eye size of flies (n = 10 per genotype) normalized to the mean of the control eye size. Expression of InR$^{Active}$ in a wild-type background with GMR driver caused eye overgrowth, while InR$^{DN}$ decreased eye size (p<0.001). Co-expression of the GR$_{36}$ with InR$^{DN}$ greatly decreased eye size (***p<0.0001), while with InR$^{Active}$ substantially increased it (two-way ANOVA followed by Holm-Sidak's multiple comparison test). Two-way ANOVA showed a significant interaction between InR genotype and expression of the repeats (p<0.0001). Data is presented as mean ± 95% confidence intervals. (**C**) Eye size (n = 20) of 2-day-old adult *Drosophila* eyes expressing dilp2, PI3K$^{CA}$, or Akt$^{CA}$ using the GMR-GAL4 driver. Co-expression of PI3K$^{CA}$ or Akt$^{CA}$ with GR$_{36}$ repeats yielded a partial rescue of the size of the eye (***p<0.0001 and *p=0.036 respectively, two-way ANOVA, followed by Holm-Sidak's multiple comparison test). Data are presented as mean ± SD. Genotypes: (**A and B**) w; GMR-Gal4/+, w; GMR-GAL4/UAS-InR$^{Active}$, w; GMR-GAL4/UAS-InR$^{DN}$, w; GMR-Gal4, UAS-GR$_{36}$/+, w; GMR-Gal4, UAS-GR$_{36}$/UAS-InR$^{Active}$, w; GMR-Gal4, UAS-GR$_{36}$/UAS-InR$^{DN}$. (**C**) w; GMR-Gal4, UAS-GR$_{36}$/+, w; GMR-Gal4, UAS-GR$_{36}$/+;UAS-dilp2/+, w/PI3K$^{CA}$; GMR-Gal4, UAS-GR$_{36}$/+, w; GMR-Gal4, UAS-GR$_{36}$/Akt$^{CA}$, w; GMR-Gal4/+, w; GMR-Gal4/+; UAS-dilp2/+, w/ PI3K$^{CA}$; GMR-Gal4/+, w; GMR-Gal4/Akt$^{CA}$.

The online version of this article includes the following source data for figure 3:

**Source data 1.** Source data pertaining to *Figure 3B*.
**Source data 2.** Excel sheet containing source data pertaining to *Figure 3C*.

because of a negative feedback system that acts to coordinate Dilp expression levels in the central nervous system (*Grönke et al., 2010*). To better understand how insulin/IGF signalling rescued the toxic effects of poly-GR, we interrogated the effect of downstream effectors of insulin signalling. We co-expressed GR36 with activated PI3K or Akt, which function downstream of the InR, and found that over-expression of either partially rescued the eye size of GR36 flies (*Figure 3C*). Together, these observations show that increased insulin/IGF signalling through the InR/PI3K/Akt pathway can rescue neurotoxicity associated with poly-GR expression.

## Activation of insulin signalling can reduce poly-GR levels in flies

We next investigated the mechanism by which increased insulin signalling reduced *C9orf72* repeat toxicity. We tested whether activation of insulin signalling could alter the level of poly-GR present in the heads of flies expressing (G4C2)36. Using a quantitative Meso Scale Discovery (MSD) immunoassay, we found that expression of InR$^{Active}$ significantly decreased poly-GR levels (*Figure 4A*).

In order to investigate whether the effect of InR$^{Active}$ on poly-GR levels occurs at the level of RAN translation, we investigated whether InR$^{Active}$ could reduce poly-GR levels in GR100 expressing flies, which generate poly-GR but do not undergo RAN translation. Neuronal expression of InR$^{Active}$ in GR100-expressing flies reduced poly-GR levels (*Figure 4B*), with a concomitant extension of lifespan (*Figure 4C*), further confirming the protective effect of increased insulin signalling. This indicates that InR$^{Active}$ is acting downstream of RAN translation to reduce poly-GR levels, and suggests that insulin/IGF signalling activation ameliorates toxicity by decreasing poly-GR levels.

## The PI3K/Akt pathway regulates DPR levels in a mammalian cell model

In order to determine whether the insulin-PI3K/AKT signalling pathway regulates DPR levels in mammalian cells, we utilized a nanoluciferase (NLuc) reporter which contains 92 seamless G4C2 repeats, which are preceded by 120 nucleotides of the endogenous human upstream sequence and followed by NLuc, lacking a start codon, in frame with poly-GR, termed 92R-NL. Thus the NLuc signal reports on RAN translated poly-GR levels. 92R-NL was co-transfected into HEK293T cells with a control plasmid expressing an ATG-driven firefly luciferase (FLuc) as a transfection efficiency and cell number control. To modulate the insulin-PI3K/Akt pathway, cells were treated for 2 days with either the pan-Akt inhibitor MK2206 (*Hirai et al., 2010*) or the PTEN inhibitor SF1670 (*Rosivatz et al., 2006*); insulin leads to the phosphorylation and activation of Akt, thus inhibiting Akt negates the effects of insulin on the PI3K/Akt pathway (*Figure 1—figure supplement 4A*). PTEN inhibition facilitates the phosphorylation and activation of Akt, thus activating the pathway (*Figure 1—figure supplement 4A*). MK2206 increased NLuc/polyGR levels (1.77-fold ± 0.54 SD, p=0.0168) (*Figure 5A*), while SF1670 decreased NLuc/polyGR levels (0.24-fold ± 0.09 SD, p=0.041) (*Figure 5B*). Therefore, consistent with our data in flies, increasing insulin signalling via Akt decreased poly-GR levels, while inhibiting the pathway increased poly-GR.

## Insulin treatment increases survival of G4C2 repeat expressing flies

Finally, we tested if treatment with insulin itself could also mitigate toxicity in flies. We injected 0.03 mg/ml insulin into fly haemolymph, equivalent to blood, at the second and seventh day post-induction of (G4C2)36 expression, and determined the effect on fly survival. Insulin treatment significantly extended lifespan in three independent cohorts of flies (*Figure 6* and *Figure 6—figure supplement 1A and B*), while modestly reducing lifespan in control, uninduced flies, again indicating a beneficial effect of insulin signalling specific to C9orf72 repeat toxicity. Higher concentrations of insulin became toxic (*Figure 6—figure supplement 1A*), indicating there is a therapeutic window within which insulin treatment is beneficial.

## Discussion

We found impairment in insulin/IGF signalling in flies expressing either G4C2 or poly-GR repeats. We showed that enhancing insulin signalling via a constitutively active insulin receptor could rescue a range of toxic phenotypes in both G4C2 and poly-GR repeat flies, by reducing poly-GR levels. This implies that altered insulin signalling is driven by poly-GR.

The insulin/IGF pathway is highly conserved between mammals and *Drosophila*. In *Drosophila*, binding of dilps to the InR results in the activation and downstream functioning of the insulin

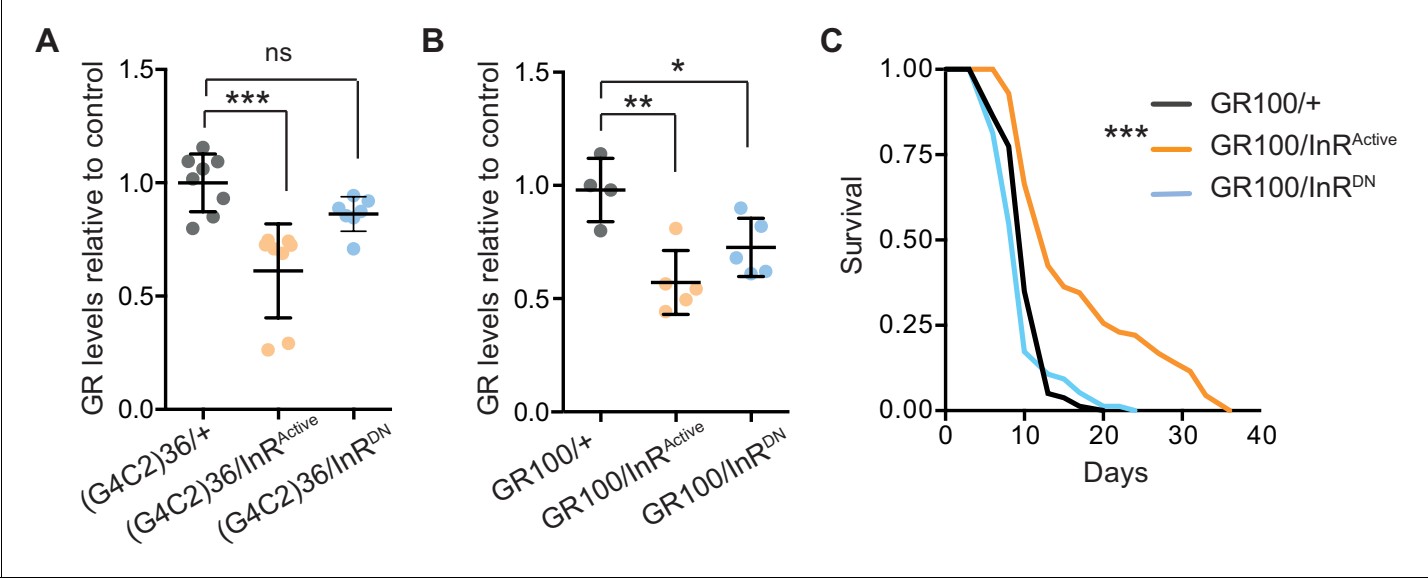

**Figure 4.** Activation of insulin signalling reduces poly-GR levels in flies. (A) GR dipeptide levels determined by Meso Scale Discovery (MSD) immunoassay were reduced in (G4C2)36 flies expressing InR$^{Active}$ compared to (G4C2)36 alone flies (***p=0.0001, one-way ANOVA, followed by Tukey's multiple comparisons test). Levels of GR were normalized to the mean GR levels of control (G4C2)36 flies. Data is presented as mean ± SD, n = 8. (B) Expression of poly-GR determined by MSD immunoassay was also reduced in flies expressing both GR100 and InR$^{Active}$ compared to flies expressing GR100 alone (**p=0.0025, one-way ANOVA followed by Tukey's multiple comparison test). Co-expression of InR$^{DN}$ slightly reduced poly-GR levels (*p=0.044). Levels of GR were normalized to the mean GR levels of control (G4C2)36. Data are presented as mean ± SD, n = 5. (C) Lifespan was significantly extended in flies (expressing ATG driven GR100 with over-expression of InR$^{Active}$ compared to flies only expressing GR100; ***p=1.62×10$^{-11}$ – log rank test). Genotypes (A) w; UAS-(G4C2)36/+; ElavGS/+, w; UAS-(G4C2)36/UAS-InR$^{Active}$; ElavGS/+, w; UAS-(G4C2)36/UAS-InR$^{DN}$; ElavGS/+. (B, C) w; UAS-GR100/+; ElavGS/+, w; UAS-GR100/UAS-InR$^{Active}$; ElavGS/+, w; UAS-GR100/UAS-InR$^{DN}$; ElavGS/+.

The online version of this article includes the following source data for figure 4:

**Source data 1.** Source data associated to *Figure 4A*.
**Source data 2.** Source data associated to *Figure 4B*.
**Source data 3.** Excel sheet containing source data pertaining to *Figure 4C*.

pathway. We found that activating this pathway through up-regulation of InR/PI3K/Akt mitigated the toxicity in the fly model, at least in part by decreasing poly-GR levels, while impairing the insulin receptor exacerbated the severity of the pathology. Importantly, this effect on poly-GR levels was confirmed in a mammalian cell model. Several studies have already shown that InR is widely expressed in the central nervous system and is involved in the regulation of diverse biological functions such as gene transcription, protein translation, and glucose transporter activity (*Boucher et al., 2014*). In a recent study, Hancock and colleagues demonstrated that InR, upon activation and nuclear transportation, associates with RNA polymerase II mainly in promoter regions of genes involved in insulin-related functions including protein synthesis, lipid metabolism, and neurodegenerative diseases (*Hancock et al., 2019*). Additionally, Minard and colleagues showed that hyperactivation of the insulin signalling pathway leads to up-regulation of the proteostatic machinery by inducing the synthesis of cytosolic chaperones (*Minard et al., 2016*). We therefore propose that overactivation of InR may improve gene transcription and translation of proteins that are crucial to DPR clearance and neuroprotection, although we cannot rule out an additional, independent effect on RAN translation in our G4C2 models.

The insulin signalling pathway plays a crucial role in regulation of growth and metabolism in neurons (*Annenkov, 2009*). Dysregulation of IGF-R signalling has been linked to a variety of neurodegenerative diseases such as Alzheimer's, Parkinson, and Huntington diseases (*Pang et al., 2016*; *Arnold et al., 2018*; *Raj and Sarkar, 2019*). However, the role of insulin signalling in *C9orf72* ALS/FTD is not yet clear. A positive correlation of incidence of ALS with early onset type 1 diabetes has

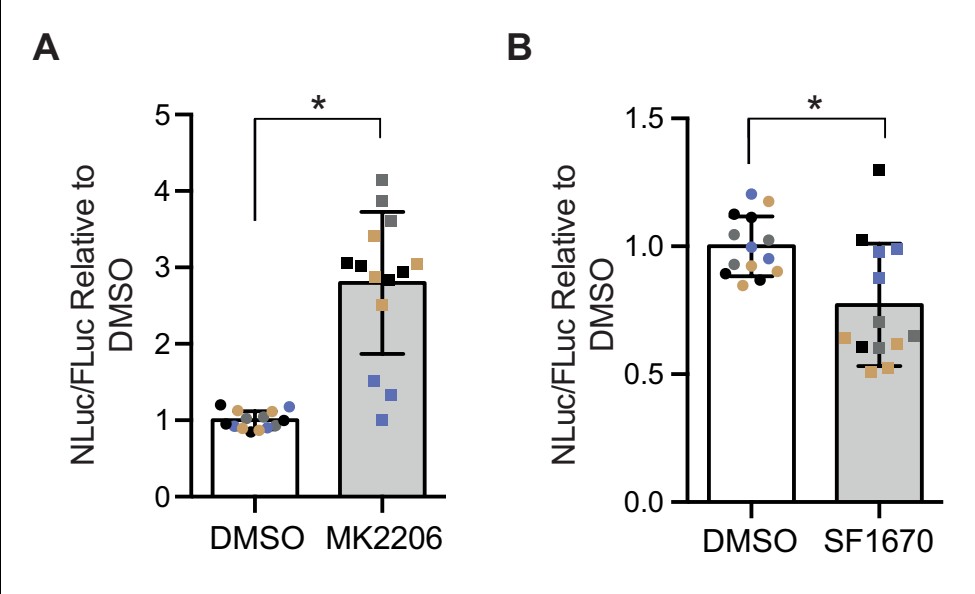

**Figure 5.** Poly-GR levels are increased by Akt inhibition and decreased by PTEN inhibition in mammalian cells. Poly-GR levels were measured using a NLuc reporter assay following a 48 hr treatment with either 1 μM MK2206 (AKT inhibitor) or SF1670 (PTEN inhibitor). (**A**) MK2206 significantly increases poly-GR levels (*p=0.0168). (**B**) SF1670 significantly decreases poly-GR levels (*p=0.0401). Each NLuc reading was normalized to FLuc for each well and further normalized to DMSO control treatment. Data given as mean ± SD of 4 biological replicates with 3–4 technical replicates per biological replicate. Data analyzed via two-tailed, unpaired Student's t-test on the mean of each biological repeat.

The online version of this article includes the following source data for figure 5:

**Source data 1.** Source data associated to *Figure 5A and B*.

been reported (*Mariosa et al., 2015*), and insulin and IGF-1 have been reported to be decreased in the blood and cerebrospinal fluid of ALS patients (*Bilic et al., 2006*), although the relevance of these findings to disease progression are unclear and confirmation in larger cohorts will be necessary. Interestingly, transcriptomic microarray analysis of *C9orf72* patient laser-capture microdissected motor neurons identified dysregulation in PI3K/Akt signalling, confirming the relevance of our findings to *C9orf72* patient material (*Stopford et al., 2017*). Reduction of Pten was also reported to reduce the toxicity of *C9orf72* repeats expressed in a mammalian cell line (*Stopford et al., 2017*), again consistent with the results we describe here, and the potential therapeutic benefit of modulating this pathway. It is also of interest that the process of brain ageing has been associated with a decrease in insulin signalling as well as impairment of insulin binding (*Zaia and Piantanelli, 1996*; *Frölich et al., 1998*), which might explain in part why ageing is a risk factor for the disease.

In recent work, *Raj and Sarkar, 2019* also identified *Drosophila* InR as a potential suppressor of poly(Q)-induced neurotoxicity and degeneration. In their study InR caused reduction of poly(Q) aggregates and improvement of the cellular transcriptional machinery. Activation of insulin signalling activation may also be implicated in the promotion of mTOR-independent autophagic clearance of poly(Q) aggregates in N2a mouse neuroblastoma cells (*Yamamoto et al., 2006*).

Interestingly, over-expression and over-activation of IGF-1R in cell tumour lines predominantly triggers activation of the RAF/MAPK and PI3K/Akt pathways, which induces proliferation and inhibits apoptosis (*Tracz et al., 2016*). In addition, IGF-1R over-expression inhibits the pro-apoptotic p53 through Akt phosphorylation (*Buck and Mulvihill, 2011*). In agreement with these reports, expression of active InR in our study resulted in decreased levels of p53 pro-apoptotic protein in diseased flies, which may attenuate neuronal apoptosis and disease progression. Since *C9orf72* repeat expansions are characterised by several altered signalling pathways (*Balendra and Isaacs, 2018*), it is

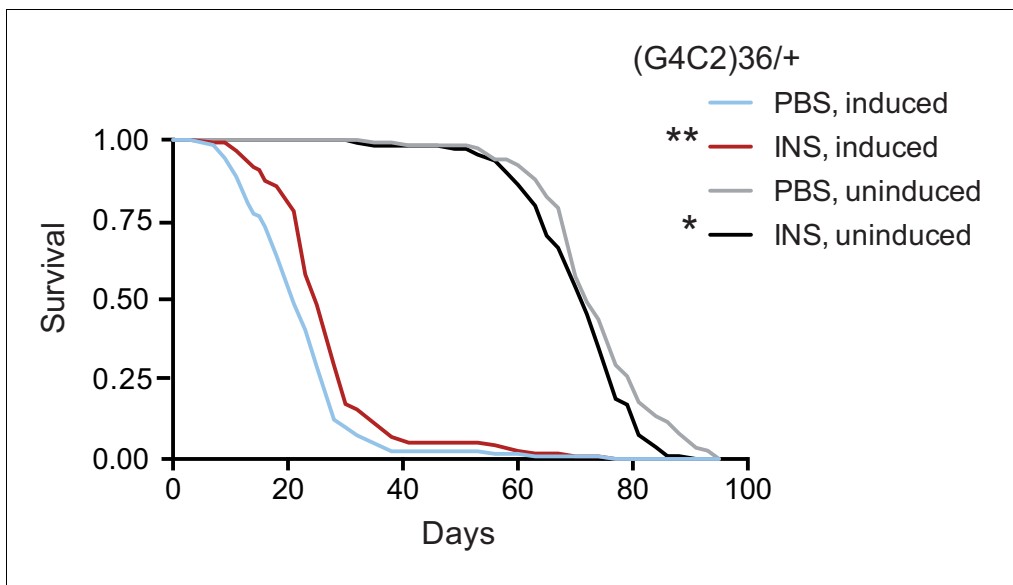

**Figure 6.** Systemic injection of insulin rescues (G4C2)36 toxicity in *Drosophila*. Injection of 0.03 mg/ml insulin (INS) significantly extended lifespan of flies (n = 120) expressing (G4C2)36 when compared with flies injected with PBS (**p=0.00034, log-rank test), while it slightly shortened lifespan in non-induced flies (*p=0.043). Genotype: w; UAS-(G4C2)36 /+; ElavGS/+.

The online version of this article includes the following source data and figure supplement(s) for figure 6:

**Source data 1.** Excel sheet containing source data pertaining to *Figure 6*.

**Figure supplement 1.** Systemic injection of insulin reduces (G4C2)36 toxicity in *Drosophila*.

**Figure supplement 1—source data 1.** Excel sheet containing source data related to *Figure 6—figure supplement 1A*.

**Figure supplement 1—source data 2.** Excel sheet containing source data associated to *Figure 6—figure supplement 1B*.

possible that increased survival of rescued flies might be a consequence of improvement of more than one molecular defect.

We found that intra-thoracic insulin administration extended survival of flies expressing G4C2 repeats. While robust, the lifespan extension was relatively modest, which might be explained by the transient nature of the insulin treatment. Insulin and IGF-1 ligands have already been tested in neuro-degenerative diseases. Intrathecal administration of IGF-1 improved motor performance, delayed the onset of disease and extended survival in the SOD1$^{G93A}$ mouse model of ALS (*Nagano et al., 2005*; *Narai et al., 2005*). However, three clinical trials of subcutaneously delivered IGF-1 in ALS reported contradictory results (*Lai et al., 1997*; *Borasio et al., 1998*; *Sorenson et al., 2008*). The contradictory outcome of these trials may have been due to insufficient drug delivery to the brain and spinal cord and the fact that ALS has heterogeneous genetic risk factors.

Overall, our study suggests that modulation of the insulin/IGF signalling pathway could be an effective therapeutic intervention against hexanucleotide repeat extension associated with *C9orf72* neurodegenerative diseases, with InR being a genetic modifier. It will be interesting in future to study the requirement for downstream effectors of insulin signalling in the toxicity rescue. Importantly, in *Drosophila* there is a single insulin-like system that has the dual function of insulin/IGF signalling; thus, the toxicity mechanism described in our model might be also related to IGFs. Therefore, it will be important to test whether insulin or IGF treatment can rescue survival in other *C9orf72* ALS/FTD vertebrate model organisms.

# Materials and methods

## Key resources table

| Reagent type (species) or resource | Designation | Source or reference | Identifiers | Additional information |
|---|---|---|---|---|
| Genetic reagent (*Drosophila melanogaster*) | Elav-GS | *Osterwalder et al., 2001* | | |
| Genetic reagent (*D. melanogaster*) | Da-GS | *Tricoire et al., 2009* | | |
| Genetic reagent (*D. melanogaster*) | GMR-GAL4 | Bloomington *Drosophila* Stock Center | BL#9146 RRID:BDSC_9146 | |
| Genetic reagent (*D. melanogaster*) | Dilp3-GAL4 | Bloomington *Drosophila* Stock Center | BL#52660 RRID:BDSC_52660 | |
| Genetic reagent (*D. melanogaster*) | UAS-(G4C2)36 | *Mizielinska et al., 2014* | | |
| Genetic reagent (*D. melanogaster*) | UAS-GR100 | *Mizielinska et al., 2014* | | |
| Genetic reagent (*D. melanogaster*) | UAS-InR$^{Active}$ | Bloomington *Drosophila* Stock Center | BL#8263 RRID:BDSC_8263 | |
| Genetic reagent (*D. melanogaster*) | UAS-InR$^{DN}$ | Bloomington *Drosophila* Stock Center | BL#8252 RRID:BDSC_8252 | |
| Genetic reagent (*D. melanogaster*) | UAS-PI3K$^{CA}$ | Bloomington *Drosophila* Stock Center | BL#25908 RRID:BDSC_25908 | |
| Genetic reagent (*D. melanogaster*) | UAS-Akt$^{CA}$ | Bloomington *Drosophila* Stock Center | BL#8194 RRID:BDSC_8194 | |
| Genetic reagent (*D. melanogaster*) | UAS-mCD8::GFP | *Lee and Luo, 1999* | | |
| Cell line (*Homo sapiens*) | HEK293T cells | UCL Drug Discovery Institute | | Mycoplasma negative HEK cells |
| Recombinant DNA reagent | pGL4.53 [luc2/PGK] Vector | Promega | #E5011 | Firefly luciferase reporter plasmid |
| Transfected construct (*H. sapiens*) | 92 repeat $G_4C_2$ nanoluciferase reporter | UCL Dementia Research Institute | | |
| Antibody | Anti-GFP (mouse, mix of two monoclonals) | Merck | Cat#11814460001 RRID:AB_390913 | WB (1:10.000) |
| Antibody | Anti-GR (rabbit) | *Moens et al., 2018* | | MSD Capture: 2 µg/ml Detection: 12 µg/ml |
| Antibody | Anti-GR (rat, monoclonal) | *Mori et al., 2013* | 5H9 | IF (1:50) |
| Antibody | Anti-dilp2 (rabbit, polyclonal) | *Okamoto et al., 2012* | | IF (1:500) |
| Antibody | Anti-non-P 4E-BP1 (rabbit monoclonal) | Cell Signalling | Cat#4923: RRID:AB_659944 | WB (1:1000) |
| Antibody | Anti-P 4E-BP1 (rabbit monoclonal) | Cell Signalling | Cat#2855 RRID:AB_560835 | WB (1:1000) |
| Antibody | Anti-p53 (mouse monoclonal) | DSHB | Dmp53-H3 RRID:AB_10804170 | WB (1:200) |
| Antibody | Anti-actin (mouse monoclonal) | Abcam | Cat#Ab8224 RRID:AB_449644 | WB (1:10.0000) |

*Continued on next page*

*Continued*

| Reagent type (species) or resource | Designation | Source or reference | Identifiers | Additional information |
|---|---|---|---|---|
| Antibody | Anti-tubulin (mouse monoclonal) | Sigma- Aldrich | Cat#T6199 RRID:AB_477583 | WB (1:2000) |
| Antibody | Anti-rat IgG-Alexa fluor 647 (goat polyclonal) | ThermoFisher | Cat#A21247 RRID:AB_141778 | IF (1:1000) |
| Antibody | Anti-rabbit IgG-Alexa fluor 488 (goat polyclonal) | ThermoFisher | Cat#A32731 RRID:AB_2633280 | IF (1:1000) |
| Antibody | HRP-conjugated anti-mouse (goat polyclonal) | Abcam | Cat#Ab6789 RRID:AB_955439 | WB (1:10.000) |
| Antibody | HRP-conjugated anti-rabbit (goat polyclonal) | Abcam | Cat#Ab6721 RRID:AB_955447 | WB (1:10.000) |
| Sequence-based reagent | Dilp2_forward | *Broughton et al., 2008* | PCR primers | ATGAGCAAGCC TTTGTCCTTC |
| Sequence-based reagent | Dilp2_reverse | *Broughton et al., 2008* | PCR primers | GACCACGGAGC AGTACTCCC |
| Sequence-based reagent | Dilp3_forward | This study | PCR primers | AGAGAACTTTGG ACCCCGTGAA |
| Sequence-based reagent | Dilp3_reverse | This study | PCR primers | TGAACCGAACTATC ACTCAACAGTCT |
| Sequence-based reagent | Dilp5_forward | This study | PCR primers | GAGGCACCTTG GGCCTATTC |
| Sequence-based reagent | Dilp5_reverse | This study | PCR primers | CATGTGGTGAGAT TCGGAGCTA |
| Sequence-based reagent | Tubulin_forward | *Moens et al., 2019* | PCR primers | TGGGCCCGTCT GGACCACAA |
| Sequence-based reagent | Tubulin_reverse | *Moens et al., 2019* | PCR primers | TCGCCGTCACC GGAGTCCAT |

## *Drosophila* stocks and maintenance

*Drosophila* stocks were maintained on SYA food (15 g/L agar, 50 g/L sugar, 100 g/L autolysed yeast, 30 ml/L nipagin [10% in ethanol], and 3 ml/L propionic acid) at 25°C in a 12 hr light/dark cycle with constant humidity. The Elav-GS stock was generously provided by Herve Tricoire (Paris Diderot University). The dilp3-Gal4 (#52660) driver and the over-expression InR constructs lines (UAS-InR$^{DN}$ #8252; UAS-InR$^{Active}$ #8263), PI3K$^{CA}$ and Akt$^{CA}$ (UAS-PI3K$^{CA}$ #25908; Akt$^{CA}$ #8194) were obtained from the Bloomington *Drosophila* Stock Centre. The GMR-Gal4 and UAS-(G4C2)36, UAS-GR100, and flies have been previously described in *Mizielinska et al., 2014*. The da-GS was kindly provided by Veronique Monnier (*Tricoire et al., 2009*), and the UAS-mCD8::GFP was a kind donation from Dr. Luo (*Lee and Luo, 1999*).

## *Drosophila* lifespan assays

The parental generation of the genotype used in each lifespan assay was allowed to lay for 24 hr on grape agar plates supplemented with yeast. Eggs were placed at a standard density into bottles containing SYA medium. Adult experimental flies were allowed to emerge and mate for 2 days before being lightly anaesthetised with $CO_2$, and females randomly allocated onto SYA containing RU486 (200 μM) at a standard density per vial (n = 15), with a minimum 150 flies per condition. Flies were tipped onto fresh food every two days and dead flies counted. Escaping flies were censored from the data.

## RNA sequencing of neuronal poly-GR100 over-expression flies

To detect differential gene expression upon adult-onset, neuron-specific over-expression of poly-GR100, flies carrying the UAS-GR100 transgene were crossed with Elav-GS driver flies. As a control, Elav-GS driver flies were crossed with wild-type flies. Experimental flies were generated as described above. Female flies were fed for three days with 200 μM RU486, and subsequently snap frozen. Total RNA was isolated from 25 fly heads using Trizol and treated with DNase. For sequencing total RNA was depleted of ribosomal RNA and libraries were generated at the Max Planck Genome Centre Cologne (Germany). This experiment was performed in triplicate. RNA sequencing was performed with an Illumina Hi-Seq2500 and 35 million single-end reads/sample and 100 bp read length at the Max-Planck Genome Centre Cologne. Raw sequence reads were quality-trimmed using Trim Galore! (v0.3.7) and aligned using Tophat2 (*Kim et al., 2013*) (v2.0.14) against the Dm6 reference genome. Multi-mapped reads were filtered using SAMtools (*Li et al., 2009*). Data visualization and analysis was performed using SeqMonk, and the following Bioconductor packages: Deseq2 (*Love et al., 2014*), topGO and org.Dm.eg.db. For visualization of functional enrichment analysis results, we further used the CellPlot package. Genes were considered to be significantly differentially expressed with an adjusted p value<0.05 and no cut-off for fold change was used. Unless stated otherwise, the set of expressed genes was used as background for all functional enrichment analyses involving expression data. The data have been deposited in NCBI's Gene Expression Omnibus (GEO) (*Edgar et al., 2002*) under the accession number GSE151826.

## RT-PCR analysis

Adult female flies were induced on SYA medium containing 200 μM R4486 for 7 days before being flash-frozen in liquid nitrogen. RNA from 12 to 15 heads per replicate was extracted using TRIzol reagent (Thermo Fisher Scientific) following the manufacturer's protocol. Approximately 1 μg of RNA per sample was treated with TURBO DNase (Thermo Fisher Scientific), followed by reverse transcription using the SuperScript II system (Invitrogen) with random hexamers (Thermo Fisher Scientific). Quantitative RT-PCR was conducted on a QuantStudio 6 Flex Real-Time PCR System (Applied Biosystems) using SYBR Green Master Mix (Applied Biosystems). Relative mRNA levels were calculated relative to alphaTub84B expression by the comparative Ct method. Primer sequences used are described in key resource table.

## Brain immunostainings

Brains from 4-day-old female flies were dissected in PBS and immediately fixed in 4% PFA in /PBS at 4°C for 2 hr and washed for 4 × 30 min in PBST (0.5% Triton X-100 in PBS). Fly brains were then blocked in PBST + 5% fetal bovine serum (FBS; Sigma #F524) for 1 hr at RT and incubated with 5H9 rat anti-poly GR (1:50) (*Mori et al., 2013*) or rabbit anti-dilp2 (*Okamoto et al., 2012*) in blocking buffer for 48 hr at 4°C. The tissues were washed 4× 30 min in PBST at RT and incubated with anti-rat IgG-Alexa Fluor 647 (ThermoFischer, catalog #A-21247) or anti-rabbit IgG-Alexa Fluor 488 (ThermoFischer, catalog #A-32731), diluted 1:1000 in blocking solution for 2 hr at RT, and washed 4× 30 min with PBST. The brains were then incubated 50% glycerol-PBS and mounted in Vectashield mounting medium (Vectorlabs, catalog #H-1200), and confocal stacks were taken with a 2 μm step size using a Leica SP8X confocal and a dry 20× (for whole brains) or a glycerol IMM 60× (for IPC zooms) objectives. The mean dilp2 immunofluorescence within each cluster of brain IPCs was calculated using the FIJI package (2.0.0-rc-43/1.51 p; NIH) software. Z projections (SUM projection) of image stacks were created. The mean fluorescence within a region adjacent to the IPCs served as background and was subtracted from the mean dilp2 fluorescence within the IPCs. Finally, a mean value representing each genotype/condition was calculated. Total numbers of IPCs were counted from each Z brain projection.

## Dot blot analysis

Total RNA of 25 female flies per genotype was extracted using Trizol and the Qiagen RNeasy Mini kit. For the dot blot analysis, 5 μg of RNA per sample were spotted onto a positively charged nylon membrane (GE Healthcare). The membrane was briefly washed with 10× SSC and RNA was then cross-linked to the membrane surface using a UVC 500 crosslinker (Amersham Biosciences). A $(GGCCCC)_5$ oligonucleotide probe was 5' labelled with γ[32P]-ATP using polynucleotide kinase to

detect sense repeats. The membrane was prehybridized with ULTRAhyb-Oligo hybridization buffer (Thermo Fisher Scientific) for 1 hr at 42°C before adding the oligonucleotide probe. Hybridisation was carried out over night at 42°C. The membrane was washed twice for 30 min in 2× SSC/0.1% SDS and then exposed to X-ray films. After autoradiography the membrane was stripped by boiling in 0.1% SDS for 30 min. For normalization the blot was re-hybridized with a probe detecting ribosomal protein RpL32 transcripts. Dot intensities were quantified in Fiji.

## Assessment of eye phenotypes

Flies carrying UAS-InR constructs, UAS-PI3K$^{CA}$ and UAS-Akt$^{CA}$ were crossed to the GMR-GAL4; UAS-(G4C2)36 driver line. The progeny were allowed to develop and eclose at 25°C; female eyes were imaged 2 days after emergence. All eye images were obtained under the same magnification; eye area was calculated from each image using ImageJ (*Schneider et al., 2012*).

## Activity and sleep analysis

Two-day-old mated female flies (n = 32) developed and eclosed under 12 hr:12 hr light:dark cycle conditions (12L:12D) were fed with food containing either 200 µM RU486 or ethanol vehicle for 12 days. After transferring into tubes, locomotor activity and sleep behaviour were recorded over 4 days in 12L:12D using the *Drosophila* Activity Monitor (DAM, TriKinetics Inc, MA) system within the experimental incubator (Percival), set at 25°C and 65% humidity. Fly activity is measured by infra-red beam crosses in the DAM tube. After 2 days of acclimatisation, data were acquired from a 24 hr period on the third day (beginning at the onset of lights-on). A custom Microsoft Excel workbook (*Chen et al., 2019*) was used to calculate total activity counts per fly in the day and night periods, and to calculate sleep minutes during the day period (continuous periods of fly inactivity lasting 5 min or longer were classified as sleep). Flies with more than 12 hr of continuous inactivity at the end of the experiment were excluded as potentially dead.

## *Drosophila* poly-GR MSD immunoassay

Heads from female flies (n = 15) induced on SYA medium containing 200 µM RU486 for 7 days were collected and processed as described previously to measure poly-GR levels (*Moens et al., 2018*).

## Fly protein extraction and western blot

Heads from female flies (n = 15) induced with 200 µM RU486 for 7 days were collected and processed as previously described (*Mizielinska et al., 2014*). Membranes were incubated overnight at 4°C with primary antibodies: mouse anti-GFP (Ab#11814460001; MilliporeSigma) (1:10.000 in TBS-T); mouse anti-actin antibody (ab8224, Abcam – 1:10.000 in TBS-T); mouse anti-tubulin (T6199, Sigma-Aldrich – 1:2000); mouse anti-p53 (dmp53-H3, DSHB – 1:200); rabbit anti-non-P 4E-BP1 (4923, Cell Signaling – 1:1000); and rabbit anti-P 4E-BP1 (2855, Cell Signaling – 1:1000). HRP-conjugated anti-mouse and anti-rabbit secondary antibodies (ab6789 and ab6721, Abcam – 1:10.000) were used for 1 hr at room temperature.

## Insulin treatment

Injections were performed twice, on the second day of repeat induction and 5 days later, by anesthetising the flies with $CO_2$. For each experiment, adult female flies (n = 80) were injected into the thorax with 32 nl of insulin (0.03 mg/ml) in PBS (pH 7.5) using a nanoinjector (Nanoject III; Drummond Scientific). Injection of the same volume of PBS acted as a control. Injected flies were then maintained at 25°C and transferred to fresh vials every third day throughout the experiment. They were collected at the indicated time points and directly processed for western blot analysis.

## Food intake – CAFE assay

In the capillary feeder assay (CAFE), a single female fly was presented with liquid food using one 10 µl calibrated capillary per chamber (n = 15 per condition). Changes in liquid meniscus height were measured over 3 days at each capillary change. Feeding volume was calculated after background subtraction of measurements from control chambers without flies.

### Nanoluciferase assay of poly-GR levels

For dual-luciferase assays, mycoplasma-free HEK293T cells were used and maintained in DMEM media supplemented with 10% FBS, 4.5 g/L glucose, 110 mg/L sodium pyruvate, and 1× GlutaMAX and kept at 37°C with 5% $CO_2$. HEK293T cells were plated at a density of 30,000 cells per well in a 96-well plate. The following day, the cells were transiently transfected with 12.5 ng of firefly luciferase expression plasmid, and 2.5 ng of RAN translated poly-GR nanoluciferase reporter plasmid (92R-NL) using Lipofectamine 2000 according to manufacturer's instructions. One hour post-transfection, cells were treated with 1 µM of either MK2206 (Cayman Chemicals, #11593), SF1670 (Merck, # SML0684), or a DMSO control. Each experiment consists of three technical replicate wells per condition, with experiments repeated three times independently. 48 hr post-transfection both firefly and nanoluciferase signals were measured using the Nano-Glo Dual Luciferase Assay according to manufacturer's instructions, on the FLUOstar Omega (BMG Labtech) with a threshold of 80% and a gain of 2000 for both readings. The nanoluciferase reading was normalised to the firefly luciferase reading for each well to control for variable transfection efficiencies and this normalised value was further normalized to the control DMSO treatment.

### Experimental design and statistical analysis

Statistical analyses were performed with Prism6 (GraphPad Software). Normality of data was tested using the D'Agostino-Pearson omnibus normality test. When data were normally distributed, a Student's t-test, one-way ANOVA, or two-way ANOVA was performed followed by multiple comparison test. For all data figures, the n values can be found in the figure legends and correspond to the number of biological repeats used in the analysis. Results were presented as mean ± 95% confidence intervals unless stated otherwise. Statistical differences were considered significant at $p<0.05$. Log-rank test on lifespan data were performed in Microsoft Excel (template available at http://piperlab.org/resources/) and data was plotted using Prism6.

## Acknowledgements

This work was funded by Alzheimer's Research UK (ARUK-PG2016A-6) (AMI), the European Research Council (ERC) under the European Union's Horizon 2020 research and innovation programme (648716 – C9ND) (AMI), the UK Dementia Research Institute (AMI), which receives its funding from UK DRI Ltd, funded by the UK Medical Research Council, Alzheimer's Society and Alzheimer's Research UK, the Wellcome Trust (LP), and the Max Planck Society (LP). We are grateful for the technical support provided by Mumtaz Ahmad, Giovanna Vinti, Enric Ureña, and Nikunj Gupta. Some stocks used in this study were obtained from the Bloomington Drosophila Stock Center (NIH P400D018537).

## Additional information

### Funding

| Funder | Grant reference number | Author |
| --- | --- | --- |
| Alzheimer's Research UK | ARUK-PG2016A-6 | Adrian M Isaacs |
| Wellcome Trust | | Linda Partridge |
| Max-Planck-Gesellschaft | Open-access funding | Linda Partridge |
| H2020 European Research Council | 648716 - C9ND | Adrian M Isaacs |
| UK Dementia Research Institute | | Adrian M Isaacs |
| Medical Research Council | | Linda Partridge |
| Alzheimer Society | | Linda Partridge |
| Alzheimer's Research UK | | Linda Partridge |

The funders had no role in study design, data collection and interpretation, or the decision to submit the work for publication.

## Author contributions
Magda L Atilano, Conceptualization, Formal analysis, Investigation, Methodology, Writing - original draft; Sebastian Grönke, Formal analysis, Supervision, Investigation, Methodology; Teresa Niccoli, Investigation, Methodology; Liam Kempthorne, Javier Morón-Oset, Oliver Hendrich, Miranda Dyson, Mirjam Lisette Adams, Idoia Glaria, Investigation; Oliver Hahn, Alexander Hull, Formal analysis, Investigation; Marie-Therese Salcher-Konrad, Amy Monaghan, Magda Bictash, Methodology; Adrian M Isaacs, Linda Partridge, Conceptualization, Supervision, Funding acquisition, Writing - original draft

## Author ORCIDs
Magda L Atilano (iD) https://orcid.org/0000-0002-3819-2023
Sebastian Grönke (iD) http://orcid.org/0000-0002-1539-5346
Idoia Glaria (iD) http://orcid.org/0000-0003-4556-489X
Adrian M Isaacs (iD) https://orcid.org/0000-0002-6820-5534
Linda Partridge (iD) https://orcid.org/0000-0001-9615-0094

## Decision letter and Author response
Decision letter https://doi.org/10.7554/eLife.58565.sa1
Author response https://doi.org/10.7554/eLife.58565.sa2

## Additional files

### Supplementary files
• Transparent reporting form

### Data availability
Sequencing data have been deposited in GEO under accession codes GSE151826. All data generated or analysed during this study are included in the manuscript.

The following dataset was generated:

| Author(s) | Year | Dataset title | Dataset URL | Database and Identifier |
|---|---|---|---|---|
| Atilano ML, Grönke S, Hahn O, Niccoli T, Kempthorne L, Morón-Oset J, Hull A, Hendrich O, Dyson M, Adams ML, Monaghan A, Salcher-Konrad MT, Bictash M, Isaacs AM, Partridge L | 2021 | mRNA profiles from heads of old female control (elavGS/+) and polyGR100 flies | https://www.ncbi.nlm.nih.gov/geo/query/acc.cgi?acc=GSE151826 | NCBI Gene Expression Omnibus, GSE151826 |

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
