## [Decision Letter]

**Acceptance summary:**

This work using appropriate *Drosophila* and mammalian cell models points to a novel therapeutic approach for C9ORF72 expansion-associated ALS/FTD patients. Key observations show that levels of insulin receptor ligands dilp2 and dilp3 are reduced in a *Drosophila* GGGGCC (G4C2) repeat expansion model and that activation of the insulin signaling mitigates multiple readouts of cytotoxicity. Together, the results suggest unusual conservation of insulin dependent neuroprotective mechanisms across species, which is of particular biological interest because *Drosophila* and mammalian insulin/ dilp secreting cells differ dramatically in anatomy and physiology.

**Decision letter after peer review:**

Thank you for submitting your article "Enhanced insulin signalling ameliorates C9orf72 hexanucleotide repeat expansion toxicity in *Drosophila*" for consideration by *eLife*. Your article has been reviewed by three peer reviewers, including Mani Ramaswami as the Reviewing Editor and Reviewer #1, and the evaluation has been overseen by K VijayRaghavan as the Senior Editor.

The reviewers have discussed the reviews with one another and the Reviewing Editor has drafted this decision to help you prepare a revised submission.

Summary:

The GGGGCC (G4C2) repeat expansion in c9orf72 is the most common genetic cause of both ALS and FTD. How expanded G4C2 repeats and their translation products such as poly(GR) induce neurodegeneration is incompletely understood. In this study, a transcriptomic analysis revealed reduced levels of mRNAs for the insulin receptor ligands dilp2 and dilp3 in flies expressing G4C2 repeats or poly(GR). Building on the observation, the authors show that activation of the insulin signaling mitigates multiple readouts of toxicity induced by poly(GR) or G4C2 repeats, possibly by decreasing poly(GR) level in these flies. Moreover, injection of insulin improved the survival of flies expressing G4C2 repeats. These findings based on studies in flies raise the possibility that the insulin signaling pathway is compromised in C9-ALS/FTD patients and that activation of this pathway may be a therapeutic approach for these disorders. If confirmed, these findings would be a timely contribution to this highly active field. However, several additional key controls and experiments are required to convincingly establish or clarify many of the above conclusions. In particular, a direct analysis of the survival of the dilp producing cells in their c9ORF model would be essential for the authors to determine whether they have discovered a cell type that shows adult degeneration in the model (which maybe valuable in itself but for reasons not addressed the current manuscript) or if the effect on insulin signalling may be a conserved feature in ALS as proposed.

Essential revisions:

There are two main lines of concern that need to be addressed before the manuscript can be accepted.

First, several controls for the genetic-interaction analyses are required to discriminate between the possibility that increased insulin signalling has a general effect on fly health or specifically improves survival and cytotoxicity in animals carrying (G4C2) repeat expansion.

Second, *Drosophila* and mammalian insulin/ dilp secreting cells appear completely different in anatomy and physiology. In *Drosophila* these cells grow very extensive and long neurites and could well be more sensitive to poly(GR) toxicity and their neurite development or even their survival could be greatly affected by elav-driven GR expression. Thus, the cellular basis for the observed dramatic decrease in insulin (Dilp2, 3) may well be quite unique to the fly model. In such a scenario, it would be difficult to extrapolate from these findings and propose a conserved pathogenic mechanism relevant to ALS patients.

1) In Figure 2, an important control is missing: the authors did not show whether activation of InR increases lifespan of control flies, such as the control (G4C2)3 flies described in an earlier report. (Not essential but interesting – considering the pro-survival effects of the insulin pathway, does InR(active) also decrease p53 level in fly models of other repeat-expansion diseases?)

2) A concern in Figure 3, as in Figure 2, is whether activation of insulin signaling promotes survival and cell proliferation in general. Thus, the effect in GR36 flies may not be specific. Indeed, expression of InR(active) or dilp2 in the wildtype eye seems to increase eye size as well; conversely, expression of InR(DN) in control flies decreases eye size (Figure 3B, C). Thus, one really cannot conclude that activation of insulin signaling "reduces" poly(GR) toxicity. Is this context p values for control flies should be stated as well. In particular, p values for comparisons between all genotypes in Panel B should be presented.

3) In Figure 5, the extent of increase in survival is so small and p value is barely <0.05 with n=80, which is at the low end. To convincingly conclude that insulin injection "rescues" (G4C2)36 toxicity, this experiment should be repeated multiple times, and in each independent experiment, a lot more flies should be examined. Moreover, it seems to be essential to demonstrate a dosage-dependent effect.

4) In Figure 1, downregulation of dilps 2, 3, and 5 in GR100 flies should be confirmed by RT-PCR as done for (G4C2)36 flies. (the sentence "in a model that expresses the pure repeats via RAN translation" should be revised. It is unclear what "pure repeats" mean.)

5) In Figure 1, the authors should determine whether the insulin-producing cells in *Drosophila* are more susceptible to poly(GR) toxicity. Does overexpression of poly(GR) preferentially kill or alter distinctive and unusual *Drosophila* insulin-producing cells? This, could explain the rather dramatic downregulation of dilps 2, 3, and 5 in GR100 flies. We suggest that the issue could potentially be addressed using genetic tools described in (https://www.nature.com/articles/mp201651).

6) Whether downregulation of insulin is a genuine pathological event in C9 patients is a major outstanding question. Some direct arguments beyond what has been provided so far would be useful. Would a collaborative experiment to measure insulin levels in published C9 BAC transgenic mice be possible to include here?

7) In Figure 4, the results seem to be too preliminary and there is no indication of what the potential mechanism for reduced GR protein level could be. The authors should at least determine whether the poly(GR) mRNA level is affected. In the Introduction, the authors state that after activation of insulin signaling, "translation initiation is increased, protein synthesis is up-regulated and autophagy is suppressed". This doesn't seem quite consistent with decreased poly(GR) level? Some more attention and explanation is needed here.

8) Using a canonically translated construct, polyGR 100X the authors find that InR CA reduces GR DPR levels, consistent with the notion that enhancing insulin signaling reduces the expression of toxic DPRs. Mechanistically, the authors place InR downstream of RAN translation based on the polyGR 100X result and the fact that InR CA mitigates G4C2 36X toxicity and also reduces polyGR generated via RAN translation. When considered in isolation, this interpretation seems reasonable however, given the authors' own conclusion that G4C2 36X levels are not changed by InR CA expression (Figure 2—figure supplement 3B) these data could also suggest that in fact increased insulin signaling inhibits RAN translation or affects the stability of poly GR produced by RAN translation.

---

## [Author Response]

Essential revisions:There are two main lines of concern that need to be addressed before the manuscript can be accepted.First, several controls for the genetic-interaction analyses are required to discriminate between the possibility that increased insulin signalling has a general effect on fly health or specifically improves survival and cytotoxicity in animals carrying (G4C2) repeat expansion.

We agree with the reviewer that is important to ascertain if increased insulin signalling is beneficial as a general effect on fly health or if it specifically reduces toxicity of the (G4C2) repeat expansion. We have therefore overexpressed InR^Active^ and InR^DN^ in the neurons of wild-type flies and observed the opposite response, namely that increased insulin signalling significantly shortens lifespan of wild type flies. This is now included in the manuscript as Figure 2B. These new data indicate that increasing insulin signalling specifically supresses C9orf72 repeat toxicity for lifespan.

Second, *Drosophila* and mammalian insulin/ dilp secreting cells appear completely different in anatomy and physiology. In *Drosophila* these cells grow very extensive and long neurites and could well be more sensitive to poly(GR) toxicity and their neurite development or even their survival could be greatly affected by elav-driven GR expression. Thus, the cellular basis for the observed dramatic decrease in insulin (Dilp2, 3) may well be quite unique to the fly model. In such a scenario, it would be difficult to extrapolate from these findings and propose a conserved pathogenic mechanism relevant to ALS patients.

This is an important point, and we have now shown that neuronal expression of poly-GR using the Elav-GS driver does not alter IPC cell number or morphology in fly brains using dilp2 immunostaining, now added as Figure 1—figure supplement 3A-C. In addition, we specifically drove poly-GR in IPC cells using a dilp3 driver and again saw no cell death – now added as Figure 1—figure supplement 3D. These new data show that expression of poly-GR is not simply killing *Drosophila* IPCs, indicating that the mechanism of dilp reduction is due to detrimental effects of poly-GR that may also be relevant in other systems.

It is noteworthy that the amino acid sequence and structure of A and B chains of the *Drosophila* insulin peptides shows a high degree of conservation with the human insulin and human insulin-like growth factors (IGFs) (Broughton and Partridge, 2009; Sajid et al., 2011). Moreover, Insulin receptor (IR) and IGF-1 receptor share higher sequence homology, and both insulin and IGF-1 are able to bind to and activate each other’s receptors to elicit activation of downstream signalling pathways e.g. PI3K/AKT and Sc-Ras-Mapk (Taniguchi, Emanuelli and Kahn, 2006; Cai et al., 2017). Importantly, human insulin can bind and activate *Drosophila* InR (Yamaguchi, Fernandez and Roth, 1995; Vinayagam et al., 2016). In mammals, insulin and IGFs are subspecialized into systems with overlapping but distinct biological function, while in *Drosophila* there is a single insulin-like system that has the dual function of Insulin/IGF signalling. The pathogenic mechanism might therefore be related to IGFs rather than insulin. Importantly, the human brain produces IGF-1 and alteration of IGF-1 levels has been implicated in neurodegeneration (Bianchi, Locatelli and Rizzi, 2017). We therefore believe that our findings in *Drosophila* are relevant and might reveal a conserved pathogenic mechanism in ALS patients involving insulin/insulin-like growth factor (IIS). We have updated the Introduction and Discussion to highlight this.

1) In Figure 2, an important control is missing: the authors did not show whether activation of InR increases lifespan of control flies, such as the control (G4C2)3 flies described in an earlier report.

We have now addressed the reviewers concern with the new data in Figure 2B, which demonstrates that increased insulin signalling extends lifespan of the diseased flies while in healthy flies it shortens lifespan. This suggests that increasing insulin signalling specifically supresses C9orf72 repeat toxicity

(Not essential but interesting – considering the pro-survival effects of the insulin pathway, does InR(active) also decrease p53 level in fly models of other repeat-expansion diseases?)

We cannot find any evidence in the literature that this has been described in fly models of other repeat-expansion diseases.

2) A concern in Figure 3, as in Figure 2, is whether activation of insulin signaling promotes survival and cell proliferation in general. Thus, the effect in GR36 flies may not be specific. Indeed, expression of InR(active) or dilp2 in the wildtype eye seems to increase eye size as well; conversely, expression of InR(DN) in control flies decreases eye size (Figure 3B, C). Thus, one really cannot conclude that activation of insulin signaling "reduces" poly(GR) toxicity. Is this context p values for control flies should be stated as well. In particular, p values for comparisons between all genotypes in Panel B should be presented.

We appreciate the reviewer comment and indeed it is well established that insulin/IGF signalling during development promotes growth of the wild type eye (Brogiolo et al., 2001). However, we observe that the alterations in the eye are significantly stronger in GR36 flies expressing active or dominant negative InR than in control flies. This indicates a further additional effect specific to poly(GR) toxicity above the effect observed in wild type flies. For example, a dramatic decrease of eye size in GR36 flies is observed when insulin signalling is further reduced by expressing InR^DN^, while the rough eye phenotype is ameliorated in the presence of an active InR. Consistent with these observations, two-way Anova analysis showed a significant interaction between repeat expression and insulin signalling modulation, with a greater effect in flies expressing GR repeats than in the control flies. As suggested by the reviewer P values for the comparisons have been added to Figure 3 and figure legend, and we have highlighted this in the Results section of the manuscript.

3) In Figure 5, the extent of increase in survival is so small and p value is barely <0.05 with n=80, which is at the low end. To convincingly conclude that insulin injection "rescues" (G4C2)36 toxicity, this experiment should be repeated multiple times, and in each independent experiment, a lot more flies should be examined. Moreover, it seems to be essential to demonstrate a dosage-dependent effect.

We have now shown that treatment with 0.03mg/ml of insulin increases survival of (G4C2)36 flies in three independent experiments – see Figure 6 and Figure 6—figure supplement 1A and B. The conclusions were consistent, with insulin significantly reducing lifespan of controls and increasing that of the disease model flies. We also administered a dose range of insulin (Figure 6—figure supplement 1A) which showed that 0.3mg/ml was not beneficial for survival of (G4C2)36 flies while 3mg/ml resulted in lethality.

4) In Figure 1, downregulation of dilps 2, 3, and 5 in GR100 flies should be confirmed by RT-PCR as done for (G4C2)36 flies. (the sentence "in a model that expresses the pure repeats via RAN translation" should be revised. It is unclear what "pure repeats" mean.)

We have performed the suggested RT-PCR experiment in GR100 flies and it showed that the transcript levels of dilp 2, 3, and 5 were significantly reduced, see Figure 1B.

5) In Figure 1, the authors should determine whether the insulin-producing cells in *Drosophila* are more susceptible to poly(GR) toxicity. Does overexpression of poly(GR) preferentially kill or alter distinctive and unusual *Drosophila* insulin-producing cells? This, could explain the rather dramatic downregulation of dilps 2, 3, and 5 in GR100 flies. We suggest that the issue could potentially be addressed using genetic tools described in (https://www.nature.com/articles/mp201651).

We have addressed this point by using some genetic tools and immunostaining described in (Monyak et al., 2017), as suggested by the reviewer. We used dilp-2 immunostaining to visualize the IPCs. We found that expression of poly-GR in the neurons using Elav-GS driver does not alter IPC cell number or morphology in the fly brains compared to control flies – Figure 1—figure supplement 3A and B. Additionally, we observed that specific expression of poly-GR repeats in IPCs using the dilp3-Gal4 driver does not induce cell death – see Figure 1—figure supplement 3D. These results, which are now included in the manuscript, argue convincingly against the idea that the reduced levels of *dilps* result from cell death of these neurons from poly-GR toxicity.

6) Whether downregulation of insulin is a genuine pathological event in C9 patients is a major outstanding question. Some direct arguments beyond what has been provided so far would be useful. Would a collaborative experiment to measure insulin levels in published C9 BAC transgenic mice be possible to include here?

We agree this is an important question. As in general the C9 BAC mice have only mild degenerative effects and it would be hard to interpret a negative result in those models. We therefore felt it was beyond the scope of the current manuscript to investigate this question in mice in a sufficiently meaningful manner.

However, given the importance of the question, we took a parallel approach and investigated whether insulin signalling affects DPR levels in mammalian cells. We used a reporter we have developed consisting of (GGGGCC)92 followed by nanoluciferase without an ATG start codon in the GR frame. Therefore, nanoluciferase signal is a direct read out of poly-GR levels. Using this reporter, we observed that, consistent with our results in *Drosophila*, activating the pathway decreases poly-GR levels while inhibiting insulin signalling increases poly-GR levels. These data are now included in the manuscript as Figure 5.

We also note that microarray data on laser capture microdissected motor neurons from C9orf72 patients showed that AKT signalling is altered (Stopford et al., 2017), which we now discuss in more detail in the Discussion.

7) In Figure 4, the results seem to be too preliminary and there is no indication of what the potential mechanism for reduced GR protein level could be. The authors should at least determine whether the poly(GR) mRNA level is affected. In the Introduction, the authors state that after activation of insulin signaling, "translation initiation is increased, protein synthesis is up-regulated and autophagy is suppressed". This doesn't seem quite consistent with decreased poly(GR) level? Some more attention and explanation is needed here.

We have now measured the mRNA levels of the repeats in (G4C2)36 flies and (G4C2)36 flies expressing InR^active^ and InR^DN^ using dot blot hybridization analysis – Figure 2—figure supplement 1B. No significant differences were observed at transcriptional level between the genotypes. The change in poly-GR levels in (G4C2)36 flies expressing InR^active^ must thus be attributable to a change in translation or degradation.

We understand the concerns of the reviewer regarding the fact that increased insulin signalling has been associated with reduced autophagy and this is potentially not consistent with decreased poly-GR levels. We have now mentioned in the Introduction and Discussion that insulin signalling pathway is a pro-survival pathway that has been associated with improved intracellular proteostasis (Minard et al., 2016) and this might play a role in poly-GR degradation.

8) Using a canonically translated construct, polyGR 100X the authors find that InR CA reduces GR DPR levels, consistent with the notion that enhancing insulin signaling reduces the expression of toxic DPRs. Mechanistically, the authors place InR downstream of RAN translation based on the polyGR 100X result and the fact that InR CA mitigates G4C2 36X toxicity and also reduces polyGR generated via RAN translation. When considered in isolation, this interpretation seems reasonable however, given the authors' own conclusion that G4C2 36X levels are not changed by InR CA expression (Figure 2—figure supplement 3B) these data could also suggest that in fact increased insulin signaling inhibits RAN translation or affects the stability of poly GR produced by RAN translation.

We agree with the reviewer that both are possible. We favour an effect downstream of RAN translation itself as in our GR model (ATG driven repeats – no RAN translation) we also observed a decrease of poly-GR. However, it is also possible that an additional effect on both RAN translation and poly-GR clearance/stability is contributing. In order to acknowledge this point, we have now added this possibility to the final sentence of the first paragraph of the Discussion.

References:

Bianchi VE, Locatelli V, Rizzi L (2017) Neurotrophic and Neuroregenerative Effects of GH/IGF1. International journal of molecular sciences 18.Cai W, Sakaguchi M, Kleinridders A, Gonzalez-Del Pino G, Dreyfuss JM, O'Neill BT, Ramirez AK, Pan H, Winnay JN, Boucher J, Eck MJ, Kahn CR (2017) Domain-dependent effects of insulin and IGF-1 receptors on signalling and gene expression. Nature communications 8:14892.Monyak RE, Emerson D, Schoenfeld BP, Zheng X, Chambers DB, Rosenfelt C, Langer S, Hinchey P, Choi CH, McDonald TV, Bolduc FV, Sehgal A, McBride SMJ, Jongens TA (2017) Insulin signaling misregulation underlies circadian and cognitive deficits in a *Drosophila* fragile X model. Molecular psychiatry 22:1140-1148.Sajid W, Kulahin N, Schluckebier G, Ribel U, Henderson HR, Tatar M, Hansen BF, Svendsen AM, Kiselyov VV, Norgaard P, Wahlund PO, Brandt J, Kohanski RA, Andersen AS, De Meyts P (2011) Structural and biological properties of the *Drosophila* insulin-like peptide 5 show evolutionary conservation. The Journal of biological chemistry 286:661-673.Taniguchi CM, Emanuelli B, Kahn CR (2006) Critical nodes in signalling pathways: insights into insulin action. Nature reviews Molecular cell biology 7:85-96.Vinayagam A, Kulkarni MM, Sopko R, Sun X, Hu Y, Nand A, Villalta C, Moghimi A, Yang X, Mohr SE, Hong P, Asara JM, Perrimon N (2016) An Integrative Analysis of the InR/PI3K/Akt Network Identifies the Dynamic Response to Insulin Signaling. Cell reports 16:3062-3074.Yamaguchi T, Fernandez R, Roth RA (1995) Comparison of the signaling abilities of the *Drosophila* and human insulin receptors in mammalian cells. Biochemistry 34:4962-4968.